# GPBP or CERT: The Roles in Autoimmunity, Cancer or Neurodegenerative Disease—A Systematic Review

**DOI:** 10.3390/ijms252313179

**Published:** 2024-12-07

**Authors:** Paula Vivó, José Miguel Hernández-Andreu, Jesús Ángel Prieto-Ruíz, Ignacio Ventura González

**Affiliations:** 1School of Medicine and Health Sciences, Catholic University of Valencia San Vicente Mártir, C/Quevedo no. 2, 46001 Valencia, Spain; paula.vivo1@estudiant.uib.cat; 2Molecular and Mitochondrial Medicine Research Group, School of Medicine and Health Sciences, Catholic University of Valencia San Vicente Mártir, C/Quevedo no. 2, 46001 Valencia, Spain; jmiguel.hernandez@ucv.es (J.M.H.-A.); jesus.prieto@ucv.es (J.Á.P.-R.); 3Translational Research Center San Alberto Magno CITSAM, Catholic University of Valencia San Vicente Mártir, C/Quevedo no. 2, 46001 Valencia, Spain

**Keywords:** GPBP, CERT, autoimmunity, ceramide, cancer, neurodegenerative diseases

## Abstract

In 1999, Goodpasture antigen-binding protein (GPBP) was identified as a protein interacting with the N-terminal region of the human Goodpasture antigen, linked to collagen IV in patients with Goodpasture syndrome, an autoimmune disease. In 2003, a splice variant lacking a serine-rich domain was discovered, which is involved in the cytosolic transport of ceramide, leading to its renaming as Ceramide Transfer Protein (CERT). This dual functionality has sparked debate regarding the roles of GPBP/CERT, as they appear to participate in distinct research fields and are implicated in various pathologies. This review follows the guidelines of the Preferred Reporting Items for Systematic Reviews (PRISMA). It compiles data from searches on Medline (PubMed) and Web of Science conducted between February and November 2022. Out of 465 records, 47 publications were selected for review. The literature predominantly focuses on GPBP/CERT as ceramide transporters. Notably, no studies contradict either hypothesis, with substantial scientific evidence supporting both roles. The need for further research is clear, and new insights into these proteins’ involvement in multiple pathologies could drive future therapeutic strategies. GPBP and CERT are multifunctional proteins with roles beyond collagen organization and ceramide transport, extending to autoimmune disorders, neurodegenerative diseases, and cancer. The ongoing controversy highlights the necessity for continued investigation, which promises to offer significant insights and potential therapeutic avenues.

## 1. Introduction

Goodpasture’s disease is an immunological disorder described only in humans and clinically characterized by rapidly progressive glomerulonephritis, occasional pulmonary hemorrhage, and the presence of autoantibodies along the glomerular and alveolar basement membranes [1,2,3]. The immune attack is carried out by antibodies directed against the α3 chain of type IV collagen [4], a major component of basement membranes. In 1999, Raya et al. [5] set out to register proteins that interact specifically with the divergent N-terminal region of the human α3NC1 domain (GP antigen) to identify possible pathogenic relevance in Goodpasture’s disease. They identified a protein known as GPBP (“Goodpasture antigen binding protein”), which is a 624-residue protein containing a high number of phosphorylatable (17.9%) and acidic (16%) amino acids, with serine being the most abundant residue (9.3%) [5]. Structurally, GPBP is a complex protein with many domains and motifs. Its expression is increased in tissues under autoimmune attack, including Goodpasture’s syndrome, lupus erythematosus, and lichen planus [2]. This suggests that GPBP may play an important role in autoimmune diseases. However, in 2003, Hanada et al. [6] described a distinct GPBP function that is completely different and unrelated to those previously described. These authors found that GPBP-2 is a key protein in the non-vesicular transport of ceramide between the ER and the trans-Golgi, where it is used for the synthesis of sphingomyelin (SM) [6] and ceramide-1-phosphate [7]. For this reason, GPBP-2 has been named CERT (ceramide transporter). The function of GPBP/CERT is controversial because it is involved in two seemingly unrelated areas of research. At present, it is unclear how the role of non-vesicular transport can be reconciled with its function in immune complex-mediated pathogenesis. Therefore, the aim of this work is to elucidate the most relevant role of the GPBP/CERT protein in autoimmunity, cancer, and neurodegenerative diseases. Recent findings highlight the role of *COL4A3BP* in tissue organization, specifically in the context of placentation. The study demonstrates that disruptions in *COL4A3BP* lead to structural abnormalities, underscoring its importance in extracellular matrix integrity. This supports the hypothesis that GPBP/CERT may play a key structural role beyond ceramide transport, potentially relevant in autoimmune and cancer pathologies [8].

Since their initial identification, extensive efforts have been made to characterize Goodpasture antigen-binding protein (GPBP) and its associated variant, ceramide transfer protein (CERT). These efforts have been aimed at elucidating their molecular properties, biological functionalities, and implications in various human pathologies. However, the definitive function of GPBP/CERT remains a subject of ongoing debate, largely due to their involvement in seemingly disparate areas of research. Otherwise, GPBP is thought to play a key role in the organization of collagen IV, a major component of basement membranes, suggesting its potential relevance in autoimmune diseases. On the other hand, some research suggests that GPBP/CERT is involved in ceramide transport, which is crucial for ceramide metabolism and sphingolipid synthesis. This dual role hypothesis positions GPBP/CERT as a subject of significant therapeutic interest in several fields, including cancer therapy, neurodegenerative diseases such as Alzheimer’s disease, and autoimmunity [9]. Therefore, a systematic review of the existing evidence on the different roles of GPBP/CERT is imperative. Such a comprehensive review will serve to establish a baseline understanding of the currently available literature and provide updated insights into the described functions of these proteins. Furthermore, it has the potential to reveal diverse therapeutic avenues and motivate further research to identify novel pharmacological targets for the prevention and treatment of the aforementioned diseases. Therefore, this systematic review not only fills an existing knowledge gap, but also provides a valuable resource for the scientific community and clinicians wishing to explore the potential clinical applications of GPBP/CERT-related research. In light of the aforementioned considerations, the primary objective of this study is to conduct a comprehensive literature review aimed at elucidating the most significant roles of the GPBP/CERT protein in relation to autoimmunity, cancer, and neurodegenerative diseases. GPBP has been implicated in the pathogenesis of autoimmune diseases, in particular Goodpasture syndrome (GPS). GPS is a rare autoimmune disease characterized by the production of autoantibodies to type IV collagen in the kidneys and lungs. GPBP has been shown to interact with type IV collagen and play a role in its organization, suggesting a potential link between GPBP and GPS.

In neurodegenerative diseases, studies have suggested that GPBP/CERT may be involved in the pathogenesis of neurodegenerative diseases such as Alzheimer’s disease and Parkinson’s disease [9]. In Alzheimer’s disease, abnormal lipid metabolism and the accumulation of amyloid beta peptides have been implicated in disease progression [10]. GPBP/CERT has been shown to interact with amyloid beta peptides and play a role in their trafficking, suggesting a potential link between GPBP/CERT and Alzheimer’s disease [11]. In cancer, GPBP/CERT has been implicated in the development and progression of several types of cancer [12], including breast cancer, liver cancer, and melanoma [13]. Several studies have shown that GPBP/CERT plays a role in the transport of ceramide to the Golgi apparatus, where it is converted to sphingomyelin, a major component of cell membranes. Abnormal sphingolipid metabolism has been implicated in the development and progression of cancer and GPBP/CERT may play a key role in this process [14].

The pathways mediated by these isoforms are illustrated in Figure 1, which provides an overview of their functional localization and interactions within the endoplasmic reticulum (ER), Golgi apparatus, and vesicular transport systems. GPBP-1 (green) is primarily associated with exocytic vesicles that deliver proteins to the extracellular space. This isoform ensures the efficient secretion of collagen type IV and other proteins into the extracellular matrix. In contrast, GPBP-2 (yellow) operates within the ER–Golgi intermediate compartment, facilitating the early stages of protein trafficking from the ER to the Golgi apparatus, a critical step in the secretory pathway. GPBP-3 (red), localized to the Golgi, participates in the processing and sorting of proteins destined for secretion or incorporation into the extracellular matrix. The figure also highlights the involvement of VAP-PP2Cε (black) as a regulator of protein retention and trafficking between the ER and Golgi compartments [13]. These molecular interactions suggest that the GPBP isoforms may have additional regulatory roles in maintaining protein homeostasis during secretion. Overall, this schematic emphasizes the multifaceted roles of GPBP isoforms in coordinating protein trafficking and secretion, particularly collagen type IV, which is a major component of the extracellular matrix. These functions are particularly relevant in autoimmune diseases, cancer, and neurodegenerative conditions, where GPBP dysregulation may have pathological implications.

Taken together, the published evidence suggests that GPBP/CERT is involved in several important functions inside and outside of the cell, including ceramide transport, collagen organization, and protein trafficking. Its involvement in various diseases highlights its potential as a therapeutic target for the development of novel treatments and therapies.

To achieve this overarching goal, we have outlined specific secondary objectives, which include describing the molecular and biochemical characterization of GPBP/CERT, providing a bibliometric analysis of the existing literature on GPBP/CERT in the context of autoimmunity, cancer, and neurodegenerative diseases, and assessing the current status of the two proposed hypotheses regarding the functions of GPBP/CERT. These objectives not only guide our research, but also ensure a holistic understanding of the relevance of GPBP/CERT in these key areas of investigation. Given the contrasting roles proposed for GPBP/CERT, this study aims to unravel the multiple functionalities of these proteins in the context of autoimmunity, cancer, and neurodegenerative diseases. The primary aim is to address two key hypotheses: Firstly, that GPBP may play a pivotal role in collagen organization, particularly in autoimmune diseases such as Goodpasture’s disease, given its increased expression in autoimmune-affected tissues and its interaction with collagen components. Secondly, we postulate that CERT’s involvement in non-vesicular ceramide transport may be important in diseases such as cancer and neurodegenerative disorders such as Alzheimer’s disease. This dual role hypothesis positions GPBP/CERT as a subject of considerable therapeutic interest across multiple fields. Through a comprehensive review of the existing literature, we aim to clarify the primary role of GPBP/CERT and provide insights that may inspire further research, potentially leading to novel therapeutic strategies.

## 2. Materials and Methods

For this systematic review following PRISMA guidelines, we conducted a comprehensive search to gather relevant scientific literature on the GPBP/CERT protein. We employed the PubMed (NCBI) and Web of Science (WOS) databases. PubMed, an open access search engine, provides access to the MEDLINE database, comprising extensive biomedical literature with MeSH (Medical Subject Headings) terms. WOS, a widely recognized and freely accessible bibliographic database, includes a vast array of academic journals, conference proceedings, and books, also indexed with MeSH terms.

To ensure the selection of pertinent scientific documents, we applied specific criteria:

Inclusion criteria:-Articles published in scientific journals, indexed with MeSH terms.-Articles written in English between 1999 and 2023.-Publications directly related to GPBP/CERT protein, tagged with relevant MeSH terms.

Exclusion criteria:-Publications lacking substantial scientific content.-Articles in languages other than English, published before 1999.-Studies unavailable as full-text articles.-Studies unrelated to the subject.-Review articles.-Articles discussing GPBP/CERT but not related to cancer, autoimmunity, or neurodegenerative diseases, not tagged with relevant MeSH terms.

We conducted the search using the MeSH terms “GPBP” and “CERT Ceramide” with the “OR” Boolean operator to ensure inclusion of potentially relevant articles.

Following PRISMA guidelines, we visually represented the search and selection process. Manual curation ensured compliance with the inclusion criteria. Using the Journal Citation Reports software and WOS data, we performed a bibliometric analysis. Data were standardized and presented as percentages to facilitate comparison across studies. Simple descriptive statistical analyses, such as calculations of means, medians, and standard deviations, were performed using Microsoft Excel. This approach ensured clarity and consistency in the presentation of results, providing a straightforward yet effective method for analyzing the data in this systematic review.

## 3. Results

Figure 2 provides a summary of the screening process and the number of articles retrieved based on the search equation and the eligibility criteria applied. A total of 465 records were identified from two databases (WOS and PubMed) using a general search equation. After applying the inclusion and exclusion criteria, 47 articles were selected for review.

Table 1 outlines the key characteristics of articles describing GPBP/CERT as a protein involved in the organization of collagen networks. One study identifies GPBP as a serine/threonine kinase, with elevated expression in tissues targeted by autoimmune responses. GPBP is also linked to the TNF-α signaling pathway, underscoring its role in inflammation. In New Zealand White mice, GPBP is associated with age-related autoimmune responses and collagen organization. Furthermore, GPBP demonstrates multi-compartmental functions, including involvement in ceramide trafficking. Human biliverdin reductase regulates both GPBP and TNF-α signaling. Additionally, T-12, an anticancer drug, disrupts EMT-based chemoresistance in solid tumors by targeting GPBP, suggesting potential for early metastasis treatment.

Table 2 presents the main characteristics of studies focused on GPBP/CERT as a ceramide transporter and its relevance to neurodegenerative diseases. The studies highlight CERT’s novel function in non-vesicular ceramide trafficking within cells. Research involving a mutant LY-A cell line, which has disrupted ceramide transport to the Golgi, identified CERT as the key factor responsible for compensating for ATP-dependent ceramide transport deficiencies between the ER and the Golgi. This finding suggests that CERTL (GPBP) is involved in ceramide transport, broadening our understanding of its cellular roles.

This research explores the basal expression of GPBP/CERT protein in the normal rat brain and its potential roles in neuroinflammation and neurodegeneration. GPBP/CERT is expressed in neuronal cells throughout the brain, with increased expression in specific regions such as the cerebral cortex, forebrain, hippocampus, and diencephalon. Additionally, novel findings show GPBP’s interaction with human serum amyloid P component, implicating it in the formation of aggregates and its colocalization within amyloid plaques in Alzheimer’s disease patients. A de novo mutation in CERT1, resulting in a gain-of-function effect, has been identified, providing insight into its role in intellectual disability. This mutation suggests a potential molecular basis for diagnostic tools and pharmaceutical interventions for intellectual disability. The study also shows that CERTL binds to Amyloid Precursor Protein (APP), influencing Aβ aggregation, neurotoxicity, and ceramide levels, opening new research avenues for Alzheimer’s disease and other neurodegenerative disorders. Furthermore, CERTL plays an immune role, particularly in modulating the pro-inflammatory status of microglia. In summary, the research sheds light on the diverse functions and implications of GPBP/CERT in brain health, including its roles in neuroinflammation, neurodegeneration, and intellectual disability.

Table 3 provides a comprehensive overview of the most relevant mutations within the CERTL gene, which encodes the CERTL protein [19]. CERTL plays a crucial role in cholesterol homeostasis and is associated with various neurological developmental disorders, such as Rett syndrome. The table is divided into three columns: “M” indicating the mutation within the CERTL gene, “C” designating the mutation cluster’s location in the protein, and “Clinical and Molecular Characteristics” offering a detailed description of the associated clinical and molecular features for each mutation. Mutations in the CERTL gene can manifest across a broad spectrum of clinical characteristics, ranging from mild developmental delays to severe neurological developmental disorders. Additionally, these mutations can influence the protein’s structure or function, with mutations affecting the protein’s structure typically resulting in more severe clinical phenotypes, potentially impacting its ability to regulate cholesterol homeostasis, while mutations impacting protein function tend to lead to milder phenotypes. This valuable information is essential for healthcare professionals when addressing patients with neurological developmental disorders linked to mutations in the CERTL gene and complements the bibliometric analysis conducted in this study.

The bibliometric analysis of the selected literature revealed that of the 47 articles, 7 focused on collagen networks (hypothesis 1), while the remaining 40 investigated ceramide transport (hypothesis 2). This indicates a predominant research focus on GPBP/CERT as a ceramide transporter. Research productivity on collagen networks remained consistently low, with a maximum of one publication per year. In contrast, publications related to ceramide transport peaked at six in 2012 and maintained a more consistent rate. Notably, there was a gap in publications on GPBP’s role in collagen networks from 2011 to 2017, whereas studies on CERT as a ceramide transporter continued consistently. Japan had the highest number of publications (16), followed by the USA (7), Spain (6), and the Netherlands and Germany (5 each). France had three publications, the UK two, and Finland, Italy, and Australia one each. It is noteworthy that the publications from Japan, Spain, and the Netherlands came from the same institutional affiliation. The most influential research groups were the ‘Centro de Investigación Príncipe Felipe’ and the ‘Instituto de Investigaciones Citológicas de Valencia’ from Spain and the ‘Department of Biochemistry and Cell Biology’ from Tokyo, Japan. The Spanish group focused on collagen organization (hypothesis 1), while the Japanese group focused on ceramide transport (hypothesis 2). In terms of topics, the highest productivity was related to ceramide transport with 22 articles, followed by neurodegenerative diseases with 8 articles, and cancer with 7. Some cancer-related articles investigated both ceramide transport and collagen network organization. The journal analysis showed that most articles were published in the first quartile of journals. *Nature* and *Cancer Cell* had the highest Journal Impact Factors and were also in the first quartile. The *Journal of Biological Chemistry* published most of the articles, with 17 out of the 47 articles selected (37.77%).

The bibliometric analysis of the selected literature reveals a significant focus on GPBP/CERT as a ceramide transporter, with only a minor emphasis on its role in collagen networks. Productivity over the years indicates a consistent interest in ceramide transport, while publications on collagen networks experienced a hiatus during certain periods. Research on GPBP/CERT is distributed worldwide, with Japan, the USA, and Spain being the most productive countries. The most influential research groups in Spain and Japan are pioneers in their respective fields. Finally, the majority of publications are in high-impact journals, underlining the importance of this research in the scientific community.

## 4. Discussion

The present study was undertaken to discuss the current state of knowledge on GPBP/CERT as responsible for the organization of collagen IV and/or non-vesicular ceramide transporter networks and thus their relevance in autoimmune pathologies, neurodegenerative diseases, and cancer.

Although the cumulative scientific knowledge on CERT as a ceramide transporter is much greater, there is sufficient evidence to support the idea that GPBP plays an important role in the organization of collagen IV and is therefore relevant to autoimmune pathogenesis: (1) GPBP phosphorylates the GP antigen at phosphorylated residues in vivo [20]. (2) Immunofluorescence-based studies have demonstrated the existence of GPBP isoforms in GBM [2,10] in addition to immunohistochemical studies showing that GPBP can be localized both extracellularly and intracellularly [2]. (3) GPBP has been reported to be overexpressed in tissues undergoing autoimmune attacks such as cutaneous lupus and lichen planus [2]. (4) Elevated levels of GPBP have been reported in association with dissociation on GBM components, deposition of IgA-like immunocomplexes, and expansion of collagen IV, demonstrating that GPBP regulates the organization of GBM collagen [10]. Moreover, no article has been found that refutes either hypothesis. It is essential to consider the ubiquitous nature of the protein, which performs one function or another depending on its location and the biomolecules with which it interacts. In any case, although the “most accepted” hypothesis is that of intracellular transporter, it is important to bear in mind that this hypothesis may be incomplete. Firstly, the CERT–ceramide transport ratio has been estimated to be 1:1 [9], but the stoichiometry is not clearly established. In the scientific literature, it is referred to as a transporter, but we suggest that biologically it may be an allosteric regulator rather than a transporter. In addition, other ceramide-metabolizing enzymes such as ceramide kinase [22] and alkaline ceramide may be localized to the Golgi apparatus, so it is possible that some of the observed phenotypes are due to other enzymes. Furthermore, there is evidence that CERT exerts its functions in cancer through the regulation of ceramide and SM. Similarly, Revert et al. [23] describe GPBP as the link between the extracellular matrix, immune response, and tumorigenesis. This study confirms that GPBP is present at high levels in the former, which could be part of the regulation of the stroma (immune response) and circulating cell (CTC) outputs. Other authors emphasize the importance of collagen IV in tumor progression [24,25,26].

According to available data, CERT levels are significantly higher in pancreatic adenocarcinoma [17] and HER2+ breast cancer [27] compared to normal tissue. On the contrary, CERT levels are significantly lower in ovarian cancer compared to normal tissue [17]. In addition, its mRNA expression is significantly reduced in human triple-negative breast cancer (TNBC) compared to non-basal tumors and normal breast tissue [28]. In addition, several lines of evidence have demonstrated the role of CERT in drug resistance. CERT is highly expressed in drug-resistant human ovarian and breast cancers [12,27,29], protecting cells from ceramide-induced apoptosis. Furthermore, *COL4A3BP* has been identified as one of the 14 genes that confer resistance to paclitaxel treatment in triple-negative breast cancer cells [12] and its expression is associated with worse outcomes in breast cancer, while low CERT expression correlates with better chemotherapeutic outcomes [27].

Taken together, the evidence suggests that CERT may be a critical factor in cancer. It is dynamically regulated and, depending on the situation, both its upregulation and downregulation can be beneficial for a tumor cell. These data confirm that cancer appears to be linked to the regulation of ceramide levels as well as to collagen and the immune system. However, further research is needed to better understand the relationship between GPBP/CERT and fundamental aspects of cancer biology, in order to control the increased autophagy of cancer cells and limit resistance to chemotherapy.

With regard to neurodegenerative diseases, sequencing analysis has identified several de novo mutations that are strongly associated with ID [30,31,32,33]. Some of these ID-associated missense mutations have been reported in the *COL4A3BP* gene. One of them is located in the codon for serine 135 in the SRM and the other one outside the SRM (G243R). Both mutations led to a defect in the hyperphosphorylation of the SRM, resulting in the expression of abnormally activated CERT. It is not known how the G243R variant is defective in SRM hyperphosphorylation.

A previous study showed that a patient with ID had a delay in myelination in the cerebrum [17]. Galactosylceramide (GalCer) is a major sphingolipid in the myelin sheath. Since ceramide is the common precursor for SM and GalCer [34], abnormally activated CERT, which preferentially delivers ceramide to the site of SM synthesis, could disrupt the proper balance between SM and GalCer in the myelin sheath. In our opinion, further studies are needed to address all these questions. Probably what is relevant in the pathological phenotype is the subcellular localization of GPBP/CERT rather than the accumulation of ceramide, which is rather the consequence of the change in localization.

On the other hand, the GPBP/CERT protein has been shown to play a relevant role in Alzheimer’s disease. Studies document that ceramide levels are increased in patients with Alzheimer’s disease [35,36,37], while SM levels are decreased [38]. The enzymes responsible for cleaving APP to generate Aβ (secretase and γ-secretase) have been found to be stabilized and have a longer half-life in Cer-enriched membranes, thereby enhancing Aβ biogenesis [39]. Thus, overexpression of CERTL restores the physiological transfer of ceramide from the ER to the Golgi, favoring SM synthesis. In addition, the reduction in ceramide levels in neuronal cells will decrease the activity of secretases and attenuate Ab formation, as shown by Crivelli et al. [18]. In addition, CERTL has been shown to bind directly to Aβ peptides and this interaction affects Aβ fibrillization by organizing Aβ into less neurotoxic aggregates [18]. Also of note is that one of the most relevant processes in the development and exacerbation of AD is neuroinflammation [40,41] Mencarelli et al. [14] demonstrated that CERTL interacts with SAP, which corresponds to the pentraxin family of the innate immune system, in addition to its ability to activate the complement system [42]. It has also been suggested that CERTL may play a role in the interaction between neurons and microglia, as the association of adeno-associated virus with CERTL affected microglial activation, despite the fact that CERTL is specifically expressed in neurons under the control of the synapsin promoter. Interestingly, neuronal-derived CERTL activity is only exerted when an ongoing inflammatory response is present, reducing membrane markers of the pro-inflammatory state of microglia [18].

Moreover, interest in CERT inhibitors has increased, although they have not yet been used clinically. Structurally ceramide-mimetic inhibitors such as HPA-12 [43] or the mycotoxin fumonisin B1, a natural inhibitor of ceramide synthases [44], and non-mimetics such as E16A [44] have been described. Limonoids have also been described to inhibit CERT via unconventional mechanisms [45]. Structurally mimetic inhibitors may be more prone to certain problems, as they may bind to both the desired target and multiple undesired targets that share the same unnatural ligand. In contrast, non-mimetics should be less likely to interfere with other proteins that share the same nature. Among the various applications of GPBP/CERT inhibitors, their therapeutic use in cancer stands out. This is because CERT inhibition leads to an increase in intracellular ceramide levels by blocking the conversion of ceramide to SM, which could be a potential strategy for cancer treatment, as well a way to increase the efficacy of existing anticancer drugs. [29]. Another important application of inhibitors would be the treatment of disorders caused by abnormally activated CERT. Murakami et al. [21] describe several mutations in CERT that are associated with intellectual disability. Therefore, in these or similar cases, the use of inhibitors could be a rational tool to reduce CERT levels to normal levels [46,47,48,49]. Finally, in addition to their physiological and pathophysiological significance, they may also prove useful in studying the interactions between GPBP/CERT and extracellular proteins such as collagen IV and the amyloid and amyloid-beta components.

Overall, the evidence presented in studies such as those by Pérez-García et al. (2018) [8] and Darris, et al. [50] provides critical support for the dual-function hypothesis of GPBP/CERT. These findings suggest that *COL4A3BP* not only serves as a ceramide transporter but also plays a pivotal role in extracellular matrix organization. Although these studies do not appear in our results table, as they were excluded based on the systematic review’s inclusion criteria, they are essential to consider as they reinforce the dual hypothesis. Disruptions in this protein’s functionality appear to compromise tissue structural integrity, particularly in pathological states such as human cancer and autoimmune diseases. The demonstrated involvement of *COL4A3BP* in placental and collagen-related structures implies that GPBP/CERT’s role in stabilizing collagen networks may be as essential as its function in ceramide transport. Thus, further investigation into GPBP/CERT as a multifunctional protein could reveal new therapeutic opportunities aimed at restoring tissue stability and targeting immune responses in these diseases. The involvement of GPBP in type IV collagen organization is crucial in maintaining basement membrane integrity, as illustrated in Figure 1. Disruptions in this process have significant implications for autoimmune and cancer pathologies, underscoring the dual-function hypothesis.

It also demonstrates the inconvenience of renaming a previously described protein. This means that all publications up to the date of the change in nomenclature will disappear in the new search. The h-index is widely accepted in the scientific community, but it does not represent 100% of the scientific impact of papers, as authors who do not publish extensively will never achieve a very high h-index. The Journal Impact Factor (JIF) of a journal is not statistically representative of its individual articles. Our inclusion criteria were carefully designed to prioritize high-relevance studies directly linked to human diseases, ensuring a comprehensive yet focused analysis. This type of metric for evaluating a specific article ends up giving more weight to the journal in which it was published than to the message contained in the article itself. It is therefore important to reflect on the fact that the dynamics of bibliometric studies and journals can overshadow results of certain scientific interest and relevance. Although our systematic review applied strict inclusion criteria focusing on human diseases, it is crucial to acknowledge that studies outside these criteria may offer additional insights into GPBP’s extracellular functions, such as its role in collagen organization and immune response.

## 5. Conclusions

Upon reviewing the scientific evidence on the role of GPBP/CERT, it is clear that the majority of the literature focuses on this protein as a ceramide transporter. This hypothesis accounts for 85.11% of the cumulative impact, compared to 14.89% based on collagen IV binding. The leading research groups in this field are located in Japan and Spain, both of which are pioneers in their respective areas of study. Additionally, Japan, the USA, and Spain have produced the most publications on this topic. Importantly, no study has refuted either hypothesis, and there is substantial scientific evidence supporting both functions. Therefore, the primary role of GPBP remains ambiguous, and further research is warranted.

Ongoing efforts should continue to investigate the relationship between GPBP/CERT and collagen IV organization, particularly its relevance to autoimmune pathogenesis. Scientific evidence also supports GPBP/CERT’s involvement in autoimmune, neurodegenerative, and cancer-related pathologies. However, our understanding in each of these areas remains incomplete. Gaining deeper insights into the complexity of this protein and its associated molecules will likely open new avenues for the development of treatments and therapies for various diseases.

Regarding the limitations of this review, it is important to note that several articles on GPBP/CERT as a ceramide transporter or related to collagen organization were excluded due to the inclusion and exclusion criteria. These exclusions were made because the articles addressed topics unrelated to cancer, autoimmune diseases, or neurodegenerative disorders.

In this study, we aimed to clarify and objectively assess the role of GPBP/CERT within the molecular biology of diseases such as cancer and neurodegenerative disorders, as well as to address the scientific implications of the nomenclature change of this protein. We evaluated the strengths, weaknesses, opportunities, and threats associated with the two proposed hypotheses.

GPBP’s role as a collagen IV organizer demonstrates several strengths: numerous studies have confirmed its association with collagen IV, particularly in autoimmune diseases such as Goodpasture syndrome. This evidence highlights its significant relevance to autoimmune pathologies, suggesting that GPBP plays a pivotal role in these conditions. Additionally, GPBP’s ubiquitous presence, both extracellularly and intracellularly, points to its broad and diverse functionality. However, some notable weaknesses remain, such as the relative scarcity of recent research on its involvement in collagen IV organization, as much of the current literature focuses on CERT as a ceramide transporter. Furthermore, the ongoing scientific debate about the primary function of GPBP introduces uncertainty, potentially diverting focus and resources in research.

Nevertheless, there are promising opportunities for the development of specific therapies. A better understanding of GPBP’s role in collagen IV organization could open new therapeutic avenues for treating autoimmune diseases. Additionally, interdisciplinary research could provide new perspectives and methods to explore the role of GPBP in various diseases. However, threats include the shift in research focus due to the predominance of studies centered on CERT, which could divert attention from GPBP’s primary function.

In contrast, the analysis of the strengths and weaknesses of the hypothesis of CERT as a ceramide transporter reveals several advantages. Most research focuses on this function, with numerous studies supporting it, and its implications in multiple diseases, including cancers and neurodegenerative disorders, underscore its clinical importance. Additionally, CERT’s relevance in drug resistance, particularly in chemotherapy for certain types of cancer, could be crucial for developing more effective treatments. However, weaknesses persist, such as the ongoing controversy over the primary function of CERT and GPBP, which generates debate over its main role. Moreover, CERT’s functional complexity, which varies depending on the cellular context and the presence of other proteins, complicates data interpretation.

Opportunities include the development of new therapeutic strategies that modulate ceramide levels to treat various diseases. Exploring additional functions of CERT could reveal new therapeutic applications and expand our understanding of its role in cell biology. However, there are also threats, such as the limited focus on CERT as a ceramide transporter, which could restrict the exploration of other potentially important functions of the protein. Studies such as those by Pérez-García et al. (2018) and Darris further support the dual functionality of *COL4A3BP*, highlighting its critical roles in both ceramide transport and structural organization of the extracellular matrix. Although not included in our systematic review results due to the exclusion criteria, these studies underscore the importance of considering GPBP/CERT’s multifunctionality in understanding its implications in cancer, autoimmune, and neurodegenerative diseases. Additionally, challenges in clinical translation due to variability in CERT’s expression and function across different cell types and tissues may hinder the development of universally effective treatments.

Future research into the molecular pathways of GPBP/CERT functions may reveal novel therapeutic strategies, particularly in regulating ceramide metabolism and collagen integrity. Targeted therapies could address aberrations in these pathways, potentially improving outcomes in diseases such as cancer and neurodegenerative disorders. To resolve the ambiguities surrounding the dual roles of GPBP/CERT, future research should prioritize multidisciplinary and integrative approaches. Structural and biophysical techniques, such as cryo-electron microscopy, could clarify the conformational states associated with ceramide transport and collagen organization. High-throughput methods like proteomics and lipidomics may provide comprehensive interaction profiles and lipid-binding patterns under diverse cellular contexts. Additionally, gene editing technologies, such as CRISPR/Cas9, offer the potential to generate specific mutants to dissect functional domains in vitro and in vivo. Collaborative efforts between biochemists, immunologists, and computational biologists could further enhance our understanding, for instance, through computational modeling integrated with experimental validation. Finally, translational studies employing patient-derived organoids or xenograft models could bridge molecular findings with clinical applications, fostering the development of targeted therapies. These experimental strategies will be essential in unraveling GPBP/CERT’s multifunctionality and its implications in health and disease.

In conclusion, this analysis highlights the strengths, weaknesses, opportunities, and threats of the two competing theories about GPBP and CERT, underscoring the need for continuous and balanced research to fully understand their roles in various pathologies. Future research must address these key areas to resolve existing controversies and harness potential therapeutic opportunities, thereby contributing to advances in the treatment of complex diseases such as cancer and neurodegenerative disorders.

## Figures and Tables

**Figure 1 ijms-25-13179-f001:**
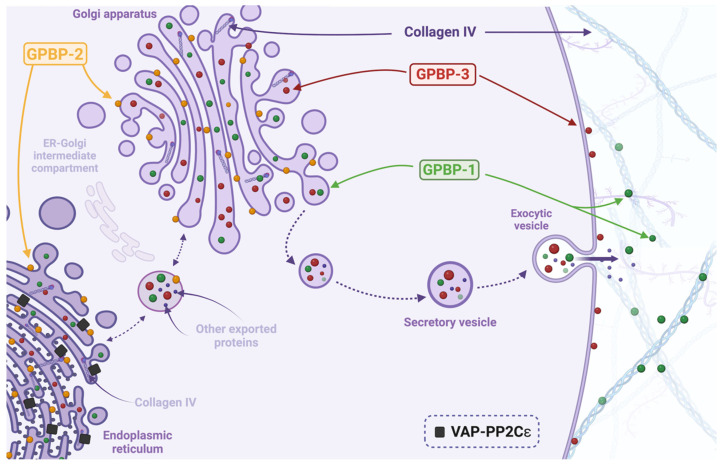
Roles of GPBP isoforms in intracellular trafficking and collagen type IV secretion. GPBP-1 (green) facilitates the exocytosis of collagen type IV and other proteins via secretory vesicles. GPBP-2 (yellow) mediates ceramide trafficking from the endoplasmic reticulum (ER) to the Golgi through the ER–Golgi intermediate compartment. GPBP-3 (red) functions within the Golgi apparatus, where it processes and sorts proteins for secretion. VAP-PP2Cε (black) regulates protein retention and trafficking within the ER–Golgi network. The figure illustrates the coordinated functions of GPBP isoforms in maintaining protein homeostasis and extracellular matrix assembly.

**Figure 2 ijms-25-13179-f002:**
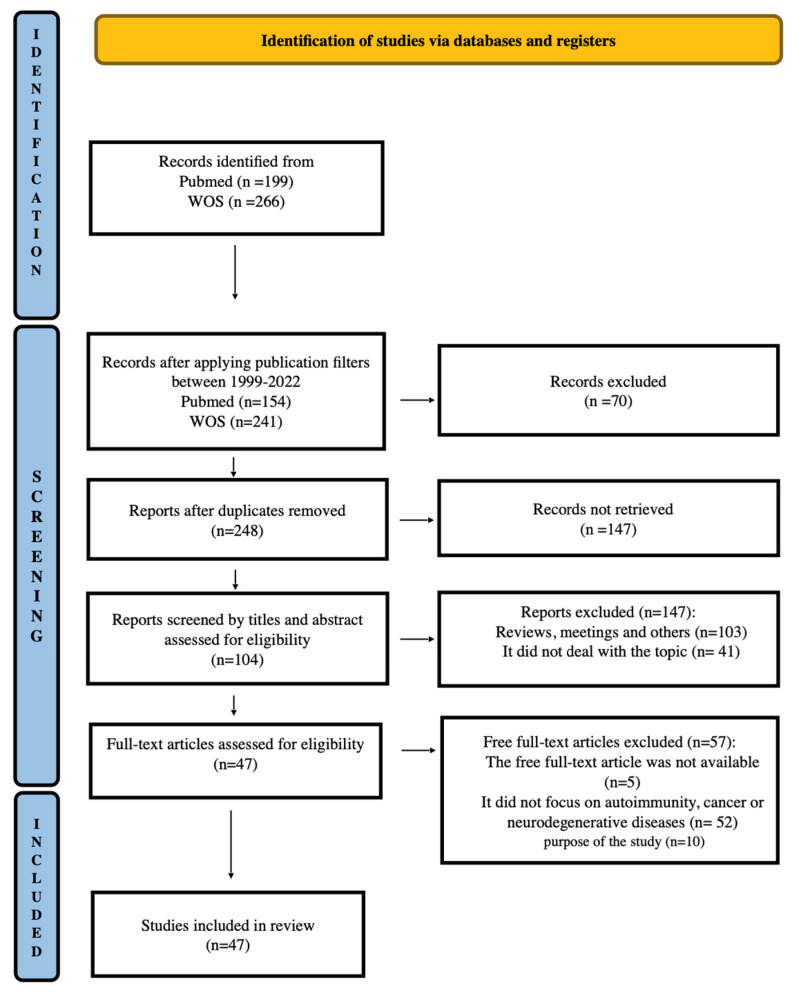
PRISMA flowchart of study selection process: the results of the PRISM-based process for literature selection illustrate the screening process and article counts resulting from our search. Initially, 465 records were identified from PubMed and WOS using a single equation. After applying chronological filters, removing duplicates, and excluding reviews and meetings, 47 articles remained that met this study’s criteria. Among the exclusions, 41 articles were unrelated to this study’s focus, while 52 addressed GPBP but not within the context of cancer, autoimmunity, or neurodegenerative diseases.

**Table 1 ijms-25-13179-t001:** GPBP/CERT as responsible for collagen IV membrane was the main feature of the studies analyzed.

Ref	Year	Summary
[5]	1999	Cloning of a previously unknown polypeptide of 624 residues: GPBP. GPBP is a novel serine/threonine kinase which, although lacking the 12 classical structural regions of a kinase, binds and phosphorylates the N-terminal region of the human GP antigen. However, its relative expression is increased in histological structures that are targets of common autoimmune responses.
[2]	2000	Report of a novel isoform of GPBP generated by alternative splicing of a 78 bp exon encoding a 26-residue serine-rich motif (GPBPΔ26). The presence of the 26-residue motif in the polypeptide chain results in a molecular species (GPBP) with preferential expression in tissue structures targeted by autoimmune responses. Thus, its expression is upregulated during autoimmune pathogenesis.
[10]	2005	This study shows that *COL4A3BP*, the gene encoding GPBP, maps head to head with POLK, the gene encoding kappa DNA polymerase. They therefore share a bidirectional promoter modulated by the necrosis factor TNF-a. Thus, the pro-inflammatory cytokine TNF-a is able to modulate transcription in the direction of the *COL4A3BP* gene, increasing mRNA levels through increased levels of the transcription factor NF-kB. This would place GPBP in the TNF-a signaling cascade.
[14]	2007	Demonstration that New Zealand White (NZW) mice develop an age-dependent autoimmune response associated with increased levels of GPBP, dissociation on GBM components, deposition of IgA-like immunocomplexes, and expansion of collagen IV. These results demonstrate that GPBP regulates the organization of GBM collagen.
[15]	2008	This work suggests that alternative exon splicing and translation initiation are strategies for targeting the products of *COL4A3BP* to multiple locations including the cytosol, secretory pathway, plasma membrane, and extracellular compartment. This demonstrates that GPBP acts in a multi-compartmental program, including phosphorylation and regulation of the molecular/supramolecular organization of proteins and inter-organic ceramide trafficking. A new 91 kDa polypeptide and a derived 120 kDa polypeptide have also been described. Both remain insoluble and associated with cellular membranes.
[16]	2010	Human biliverdin reductase (hBVR) is a regulator of the TNF-a GPBP collagen type IV signaling cascade. hBVR has an inhibitory effect on the kinase activity of GPBP and also plays a regulatory role in the response of GPBP to TNF-a and its transcriptional regulation by NF-kB.
[17]	2018	Demonstration that T-12 specifically targets mesenchymal GPBP and disrupts EMT-based chemoresistance in solids, including lung and breast cancer. T-12 is a first-in-class anticancer drug candidate that selectively targets the collagen of the cell microenvironment. This will allow the motorization of GPBP levels and the selection of patients who will respond to this drug, thus predicting metastasis formation for early treatment.

**Table 2 ijms-25-13179-t002:** Main characteristics of studies analyzed focused on GPBP/CERT as a ceramide transporter and its relevance in neurodegenerative diseases.

Ref	Year	Summary
[6]	2003	A new function for GPBPΔ26 (CERT) was reported, focusing on the intracellular trafficking of ceramide in a non-vesicular manner. Analysis of a mutant mammalian LY-A cell line, in which the ceramide transport pathway to the Golgi is altered, identified CERT as the defective factor. These results indicate that the expression of CERT fully compensates for the deficiency in the ATP-dependent pathway of ceramide transport from the ER to the Golgi in LY-A cells. In addition, this work suggests that CERTL (GPBP) is also functional in ceramide transport.
[18]	2009	Study of the basal expression of GPBP/CERT protein in normal rat brain to analyze the possible function of GPBP in neuroinflammation and neurodegeneration. It was shown that GPBP/CERT is present in neuronal cells and is widely distributed throughout the brain, with increased expression in some regions of the cerebral cortex, forebrain, hippocampus, and diencephalon.
[19]	2012	This study aims to test whether CERT levels are altered in the acute neurodegenerative process. An experimental 6—ODHA model was used to imitate dopamine depletion (rats chronically dopamine (DA) depleted by bilateral striatal injections of 6-hydroxydopamine) as an animal model of Parkinson’s disease (PD). No differences in GPBP/CERT expression levels were detected between diseased and control animals. This suggests that the expression pattern of GPBP/CERT in the striatum is not affected in the 6-OHDA rat model of PD.
[20]	2012	Novel evidence that GPBP binds to human serum amyloid P component (SAP), a non-fibrillar glycoprotein belonging to the pentraxin family of innate immune proteins. SAP and GPBP form aggregates in the blood and some colocalize in the amyloid plaques of Alzheimer’s disease patients.
[21]	2020	Identification of a novel de novo nonsense mutation replacing a serine with a proline at position 135 in CERT1 (S135P) in a patient with severe intellectual disability and systemic symptoms. This mutation induced a gain-of-function effect on its activity. Biochemical analysis showed that S135 is essential for the hyperphosphorylation of a serine repeat motif of CERT, which is required for the downregulation of CERT activity. These results provide a potential molecular basis for new diagnostics and a potential pharmaceutical intervention for intellectual disability caused by gain-of-function mutations in CERT1.
[9]	2021	The results of this study show that CERTL binds to APP, modifies Aβ aggregation, and reduces Aβ neurotoxicity in vitro. In addition, increasing CERTL modulates SL levels (by reducing specific ceramide and increasing SM) and affects amyloid plaque formation and brain inflammation in AD. This opens up research avenues for therapeutic targets in Alzheimer’s and other neurodegenerative diseases. In addition, a novel immune role for CERTL is reported, as an adeno-associated virus decreased membrane marker important for the pro-inflammatory status of microglia.
[22]	2021	Demonstration that several CERT variants with intellectual disability (ID)-associated mutations frequently impair serine repeat motif (SRM) phosphorylation-dependent repression, resulting in increased SM synthesis and concomitant subcellular redistribution of CERT. The mutations were a serine-to-leucine substitution in the SRM (S132L mutation) and a glycine-to-arginine substitution outside the SRM (G243R mutation) in CERT.

**Table 3 ijms-25-13179-t003:** Phenotype–genotype correlations and main molecular characteristics of the most relevant CERTL mutations.

C	M	Clinical Features	Molecular Features
2	p.S132 (*n* = 4)	Individuals born significantly underweight, developmental delay before the first year of life, and failure to thrive (may become immobile by late adolescence). Severe seizures have been reported. Greatest motor delay. ID profound to severe. One of the most severe mutations.	Cluster 1 mutations have a greater tendency than WT to be associated with perinuclear membranes and cytosolic dots. In terms of phosphorylation patterns, cluster 1 showed hypophosphorylation. These mutations did not affect CERT phosphorylation outside the SRR or SRR monophosphorylation, suggesting that CSNK1G2-mediated SRR phosphorylation is defective in these mutants. In addition, CERT expression in cluster 1 mutants increased dhCer and dhSM to a greater extent than in WT. p.S132 and p.S135 were associated with the largest increases. In addition, twice as much SA was detected compared to WT CERT.
3	p.G243 (*n* = 3)	Normal birth weight, no perinatal problems and early developmental milestones, but then regression or slowed development (except for one subject who was identified in infancy). Speech delay, which may lead to non-verbalism. Seizures reported. Severe ID. One subject was noted to have behavioral problems consistent with ASD.	Greater tendency than WT to be associated with perinuclear membranes and cytosolic dots. (A similar phenotype was observed in patient-derived fibroblasts carrying the p.G243R mutation.) In terms of phosphorylation patterns, they showed hypophosphorylation. These mutations did not affect CERT phosphorylation outside SRR or SRR monophosphorylation, suggesting that CSNK1G2-mediated SRR phosphorylation is defective in these mutants. CERT overexpression in cluster 3 mutants increased dhCer and dhSM to a greater extent than in WT. p.G243 was associated with the greatest increases. In addition, twice as much SA was detected compared to WT CERT. These variants are located in regions involved in secondary structure formation and/or protein–protein interactions. In addition, cluster 3 variants affect CERT regulation downstream of PKD phosphorylation.
	p.G243 (*n* = 3)	Born slightly underweight with hypotonia and feeding difficulties. No failure to thrive. No verbal ability. No history of seizures. Neurosensory problems and severe ID.	
	p.T251 (*n* = 1)	Growth retardation with musculoskeletal problems and hypotonia. Severe speech delay. No seizures reported. Severe ID	
**C**	**M**	**Clinical Features**	**Molecular Features**
4	p.V326 (*n* = 1)	Born slightly underweight with mild motor delay. Speech delay and moderate ID. Behavioral problems consistent with ASD.	Cluster 4 variants showed a greater tendency than WT to be associated with perinuclear membranes and cytosolic dots. These mutations did not show hypophosphorylation. Overexpression of CERT in cluster 3 mutants increased dhCer and dhSM to a greater extent than in WT. In addition, twice as much SA was detected compared to WT CERT.
	p.A329 (*n* = 1)	Born slightly underweight with hypotonia and failure to thrive. Non-verbal. Seizures have been reported. Neurosensory problems and severe ID. Behavioral problems consistent with ASD.	
	p.L330 (*n* = 2)	Individuals born significantly underweight, hypotonic and with developmental delay. Episodic seizures. Neurosensory problems and severe ID. One subject was noted with behavioral problems consistent with ASD.	

## Data Availability

No new data were created or analyzed in this study.

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
