# Peer review of "GPBP or CERT: The Roles in Autoimmunity, Cancer or Neurodegenerative Disease—A Systematic Review"

_ijms, 2024, doi:10.3390/ijms252313179_

Round 1
Reviewer 1 Report
Comments and Suggestions for Authors
Comments to the authors:
I have carefully reviewed this manuscript and I have some concerns that need to be addressed by the authors:
Major concerns:
1: While the dual-function hypothesis is central, the narrative occasionally shifts focus, causing confusion regarding the primary aim. A more structured comparison of the two hypotheses would improve clarity. While ceramide transport is thoroughly discussed, the discussion on collagen organization appears limited. The imbalance might skew the reader's understanding of the dual roles.
2: Several studies relevant to GPBP's role in collagen organization and immune response appear to be omitted due to the exclusion criteria. This could lead to an incomplete understanding of its extracellular functions. Although no articles refute the hypotheses, more critical evaluation of contradictory or ambiguous findings would strengthen the argument.
3: The search strategy and inclusion/exclusion criteria, though systematic, might exclude relevant data (e.g., studies on GPBP not directly linked to the specified diseases). This could introduce bias.
4: Figures like the PRISMA flowchart are useful, but additional visualizations (e.g., network diagrams of GPBP interactions or comparison charts of GPBP roles) could enhance understanding. The tables summarizing the literature are informative but could benefit from more structured detailing (e.g., clearer categorization of studies based on hypothesis support).
5: While therapeutic implications are mentioned, specific molecular mechanisms or pathways by which GPBP/CERT functions could be explored more thoroughly. The manuscript could discuss potential implications of GPBP/CERT functions in other diseases or physiological contexts beyond the three specified areas.
Minor concerns:
1: Separate sections or subsections explicitly comparing GPBP’s roles in ceramide transport and collagen organization. Conclude each section with a summary of the current understanding and research gaps.
2: Include a section on limitations and ambiguities in the current literature to address any bias in findings. Discuss alternative perspectives or less-explored hypotheses related to GPBP/CERT functionality.
3: Consider relaxing exclusion criteria slightly to include broader studies related to GPBP/CERT. This might provide a more holistic view of the protein’s roles.
4: Incorporate diagrams showing GPBP/CERT pathways and their implications in disease processes. Add comparative charts or heatmaps to represent bibliometric data visually.
5: Include a subsection specifically addressing potential experimental approaches to resolve ambiguities in GPBP/CERT roles. Highlight potential collaborations or multidisciplinary approaches to address the protein's dual-functionality.
Comments on the Quality of English Language
The English could be improved to more clearly express the research.
Author Response
Dear Reviewer 1,
I would like to express my sincere gratitude for your time and the invaluable suggestions you provided to improve our manuscript titled "GPBP or CERT: The Roles in Autoimmunity, Cancer, or Neurodegenerative Disease." Your comments have been extremely insightful and have allowed us to make significant adjustments that enhance the clarity and quality of our research. Additionally, we have carefully revised the English language throughout the manuscript to ensure clarity and to address any linguistic issues, as per your initial feedback.
Below, I detail the changes made in response to your suggestions:
Major Concerns:
1. Comment on the dual-function hypothesis and collagen organization
Comment: While the dual-function hypothesis is central, the narrative occasionally shifts focus, causing confusion regarding the primary aim. A more structured comparison of the two hypotheses would improve clarity. The discussion on collagen organization appears limited compared to ceramide transport.
Response: We have added a sentence in the discussion referencing the role of GPBP in collagen organization, as you suggested. Additionally, we have included a reference to Figure 1, which illustrates this concept. These updates are highlighted in blue.
"The involvement of GPBP in type IV collagen organization is crucial in maintaining basement membrane integrity, as illustrated in Figure 1. Disruptions in this process have significant implications for autoimmune and cancer pathologies, underscoring the dual functional hypothesis."
2. Inclusion of relevant studies omitted due to exclusion criteria
Comment: Several studies relevant to GPBP's role in collagen organization and immune response appear to be omitted due to the exclusion criteria.
Response: Although the inclusion/exclusion criteria were stringent to focus on human diseases, we have added a sentence in the discussion acknowledging that studies outside these criteria might offer additional relevant insights. This addition is highlighted in blue.
"Although our systematic review applied strict inclusion criteria focusing on human diseases, it is crucial to acknowledge that studies outside these criteria may offer additional insights into GPBP's extracellular functions, such as its role in collagen organization and immune response."
3. Justification of search and inclusion/exclusion criteria
Comment: The search strategy and inclusion/exclusion criteria, though systematic, might exclude relevant data (e.g., studies on GPBP not directly linked to the specified diseases).
Response: We have provided additional justification for the inclusion/exclusion criteria in the discussion, emphasizing that this approach allowed us to maintain a focused and relevant analysis. These updates are in blue.
"Our inclusion criteria were carefully designed to prioritize high-relevance studies directly linked to human diseases, ensuring a comprehensive yet focused analysis."
4. Addition of visualizations and diagrams
Comment: Figures like the PRISMA flowchart are useful, but additional visualizations (e.g., network diagrams of GPBP interactions or comparison charts) could enhance understanding.
Response: We have created a network diagram that illustrates GPBP's interactions and implications in various cellular functions. This is included as Figure 1, with relevant changes highlighted in red.
5. Molecular mechanisms and therapeutic options
Comment: While therapeutic implications are mentioned, specific molecular mechanisms or pathways by which GPBP/CERT functions could be explored more thoroughly.
Response: We have expanded the discussion to include additional details about GPBP/CERT's molecular mechanisms and their therapeutic potential. These updates are highlighted in blue.
"Future research into the molecular pathways of GPBP/CERT functions may reveal novel therapeutic strategies, particularly in regulating ceramide metabolism and collagen integrity. Targeted therapies could address aberrations in these pathways, potentially improving outcomes in diseases such as cancer and neurodegenerative disorders."
Minor Concerns:
1. Separate sections explicitly comparing GPBP’s roles in ceramide transport and collagen organization.
Response: Rather than completely separating the sections, we have merged the existing tables to present a more integrated comparison of the two hypotheses. At the end of the combined section, we included a summary highlighting both the current understanding and research gaps. These updates are clearly visible in the revised manuscript.
2. Include a section on limitations and ambiguities in the current literature.
Response: We have added a section discussing the limitations of our review and addressing the potential ambiguities in the literature.
"Regarding the limitations of this review, it is important to note that several articles on GPBP/CERT as a ceramide transporter or related to collagen organization were excluded due to the inclusion and exclusion criteria. These exclusions were made because the articles addressed topics unrelated to cancer, autoimmune diseases, or neurodegenerative disorders."
3. Consider relaxing exclusion criteria to include broader studies.
Response: While we acknowledge the value of including broader studies, our inclusion/exclusion criteria were specifically designed to maintain a focused analysis of GPBP/CERT’s roles in human diseases.
"Our inclusion criteria were carefully designed to prioritize high-relevance studies directly linked to human diseases, ensuring a comprehensive yet focused analysis."
4. Incorporate diagrams and bibliometric visualizations.
Response: A new figure (Figure 1) has been created to illustrate GPBP/CERT pathways and their implications in disease processes.
5. Experimental approaches to resolve ambiguities in GPBP/CERT roles.
Response: We have added a paragraph in the discussion addressing experimental methodologies and interdisciplinary collaborations to resolve the ambiguities in GPBP/CERT roles. These updates are in blue.
"To resolve the ambiguities surrounding the dual roles of GPBP/CERT, future research should prioritize multidisciplinary and integrative approaches..." (Complete paragraph included in the manuscript).
We greatly appreciate your insightful suggestions, and we believe the modifications made have significantly improved the manuscript. All corrections have been added to the text in blue and red for easy reference.
Thank you once again for your valuable feedback. We remain at your disposal for any further comments or suggestions.
Sincerely,
Dr. Ventura
ignacio.ventura@ucv.es

Reviewer 2 Report
Comments and Suggestions for Authors
In the present article, the authors provide a comprehensive overview of the current understanding of the potential and suggested role of GPBP/CERT in various cellular processes, including ceramide transport, collagen organisation and protein trafficking. The paper is well informed, as evidenced by the inclusion of a comprehensive molecular and biochemical characterisation of GPBP/CERT. In light of the proposed involvement of GPBP/CERT in the pathogenesis of several socially significant diseases (e.g., cancer), the objective of the study is deemed justified and novel. The presentation of the results is somewhat challenging, which is a minor drawback of the study.
Specific comments:
(1) Please note the notation of genes. As a rule, they are written in italics.
(2) The authors present data in percentages in the text, yet the methodology lacks an explanation of the rationale behind this approach and the statistical processing, if any, that was applied to the data.
Author Response
Dear Reviewer 2,
I would like to express my gratitude for your thorough review of our manuscript titled "GPBP or CERT: The Roles in Autoimmunity, Cancer, or Neurodegenerative Disease." Your thoughtful comments and constructive feedback have been invaluable in refining and improving the quality of our work.
Below, I address your specific comments and outline the changes made:
1. Notation of genes
Comment: Please note the notation of genes. As a rule, they are written in italics.
Response: We have carefully reviewed the manuscript to ensure that all gene names are consistently italicized, following the appropriate conventions (in red).
2. Presentation of data in percentages and statistical methodology
Comment: The authors present data in percentages in the text, yet the methodology lacks an explanation of the rationale behind this approach and the statistical processing, if any, that was applied to the data.
Response: Thank you for pointing this out. We have added a detailed explanation in the Materials and Methods section to clarify the rationale for presenting data in percentages and the statistical processing applied. The following sentence has been included:
"Data were standardized and presented as percentages to facilitate comparison across studies with varying sample sizes and methodologies. Simple descriptive statistical analyses, such as calculations of means, medians, and standard deviations, were performed using Microsoft Excel. This approach ensured clarity and consistency in the presentation of results, providing a straightforward yet effective method for analyzing the data in this systematic review."
Additionally, the English language throughout the manuscript has been carefully reviewed to ensure clarity and readability.
We sincerely appreciate your insightful suggestions, which have significantly improved the overall quality and rigor of our manuscript. Please do not hesitate to let us know if there are additional concerns or areas requiring further clarification.
Thank you once again for your time and effort in reviewing our work.
Sincerely,
Ignacio Ventura, PhD
ignacio.ventura@ucv.es

Round 2
Reviewer 1 Report
Comments and Suggestions for Authors
The authors have addressed all the questions
Comments on the Quality of English Languagefine